# AutoPlan: Automatic Planning of Interactive Decision-Making Tasks With Large Language Models

**Siqi Ouyang**
Computer Science Department
University of California Santa Barbara
siqiouyang@ucsb.edu

**Lei Li**
Language Technology Institute
Carnegie Mellon University
leili@cs.cmu.edu

## Abstract

Recent large language models (LLMs) are promising for making decisions in grounded environments. However, LLMs frequently fail in complex decision-making tasks due to the misalignment between the pre-trained knowledge in LLMs and the actual rules in the environment. Existing methods require either costly gradient computation or lengthy in-context demonstrations. In this paper, we propose AutoPlan, an approach to guide LLM-based agents to accomplish interactive decision-making tasks. AutoPlan augments the LLM prompt with a task-solving plan and optimizes it through iterative experience collection and reflection. Our experiments show that Auto-Plan, though using no in-context demonstrations, achieves success rates on par with the baselines using human-written demonstrations on ALFWorld and even outperforms them by 8% on HotpotQA. The code is available at https://github.com/owaski/AutoPlan.

## 1 Introduction

The ability to make decisions lies at the core of human intelligence, enabling us to navigate through a multitude of choices and select the best possible actions based on available information. Recent large language models, trained with trillions of tokens, have gained impressive reasoning ability and now have the potential to act as autonomous agents for decision-making tasks in grounded environments (Zhang et al., 2022; Chowdhery et al., 2022; OpenAI, 2023; Touvron et al., 2023).

Decision-making tasks in grounded environments can be as simple as calculating mathematical problems with an external calculator or as complex as doing housework. Current LLM can easily use an external calculator by decomposing the formula into atomic function calls (Bubeck et al., 2023). However, LLMs frequently fail in more complex tasks in an environment with many objects and prerequisite dependencies. Considering the *Heat* task

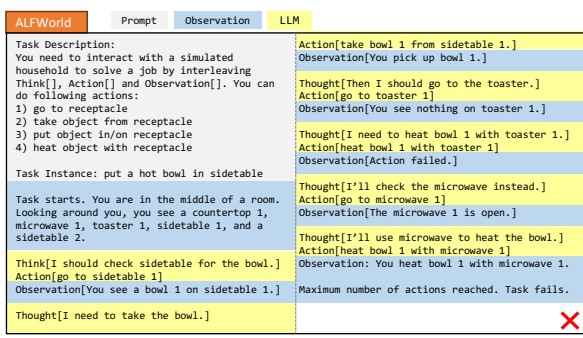

Figure 1: Problem illustration: Planning for decision-making tasks. Given the description of an environment, legit actions, and a task instance, the goal is to guide an LLM to generate a sequence of thoughts and actions (highlighted in yellow) reacting to observations provided by the environment (highlighted in blue) to solve the task instance. An LLM may fail in complex tasks due to the lack of prior knowledge.

in ALFWorld (Shridhar et al., 2021)), LLM agents struggle to find the correct action sequence within the maximum number of actions (Figure 1). The primary reason behind such failures is the misalignment between LLM's pre-trained knowledge (e.g., generating fluent sentences) and the concrete rule of the grounded environment (e.g., household item functionality in ALFworld). In the ALFWorld environment, the agent can only heat an object with a microwave instead of a toaster. However, the LLM does not learn such knowledge during pretraining, eventually failing the task.

Existing methods aligning LLMs to desired environments either employ reinforcement learning (RL) and imitation learning (IL) methods (Ouyang et al., 2022; Carta et al., 2023), or provide a few demonstrations to conduct in-context learning (ICL) (Yao et al., 2023). On the one hand, RL and IL methods require computationally costly gradient computation for existing LLMs. On the other hand, the performance of ICL methods highly depends on the selection of demonstrations.

In this work, we propose AutoPlan, a purely

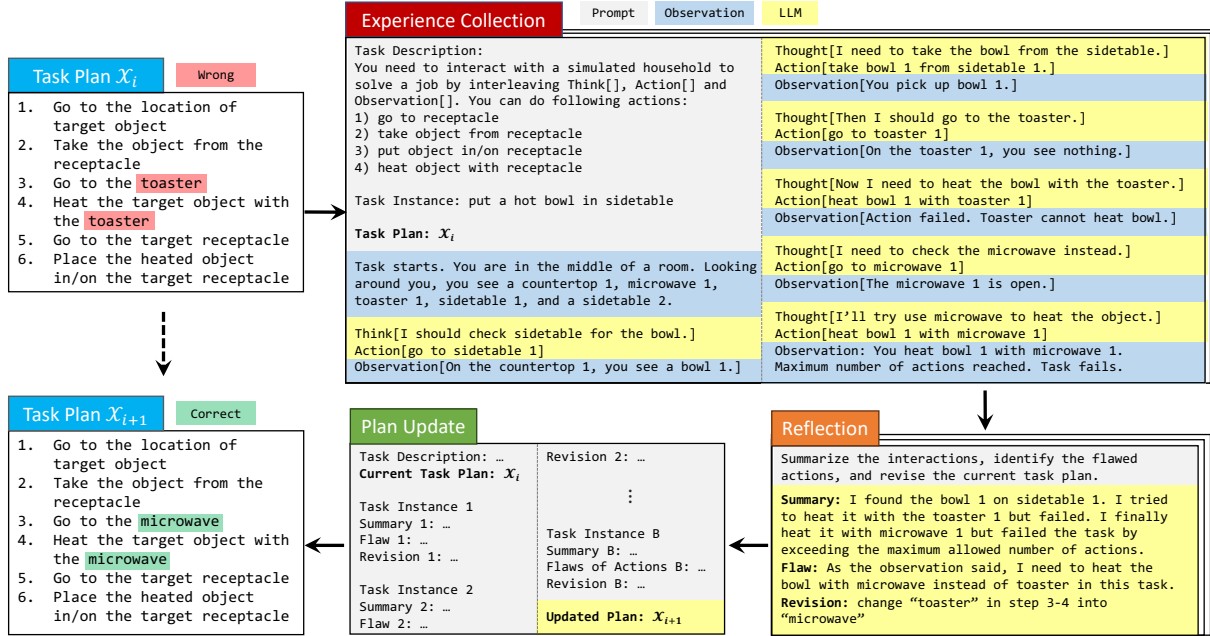

Figure 2: One optimization iteration of AutoPlan on *Heat* task of ALFWorld. Given the current plan $\mathcal{X}_i$ (with wrong steps highlighted in red), the LLM agent collects interaction experiences from a batch of task instances (prompts and LLM outputs are highlighted in grey and yellow, respectively). Then, the agent reflects on the experiences and outcomes through summarization, flaw identification , and plan revision. Finally, the agent aggregates the current batch of task instances together with their reflections and updates the task plan to $\mathcal{X}_{i+1}$ (with correct steps highlighted in green).

prompt-based method, to guide an LLM to solve such decision-making tasks without costly gradient computation or in-context demonstrations. In high-level speaking, AutoPlan solves the task by iteratively interacting with the given environment conditioning on a task plan described in natural language. Figure 2 illustrates how AutoPlan finds an optimal plan to guide the LLM to heat an object correctly and put it at the target location. AutoPlan starts with an empty plan and uses the LLM agent to collect interaction experiences conditioning on an initial incorrect plan. Then AutoPlan instructs the LLM agent to reflect on the collected experiences and revise the task plan based on the reflection. It further deploys the new task plan to collect more experiences with the LLM agent.

The primary technical challenge of this approach is to ensure stable and progressive plan optimization since the plan expressed in natural language can be highly slapdash and versatile. We propose two techniques in AutoPlan: (1) experience batching and (2) SIR reflection. We batch multiple experiences before updating the plan to help reduce variance. We introduce an explicit SIR reflection (Summarization, flaw Identification, plan Revision) to elicit helpful information from the interaction experience. We evaluate AutoPlan and other methods on two distinct benchmarks.

Our contributions are:

- We propose AutoPlan, a novel prompting method to align LLMs with the need for grounded decision-making tasks without computing gradients or using human-written demonstrations.

- Our experiments show that AutoPlan achieves on-par success rates with baselines involving human-written demonstrations on ALFworld (Shridhar et al., 2021) and even 8% higher accuracy on HotpotQA (Yang et al., 2018).

- We verify that larger batch size leads to more stable learning, and the explicit SIR reflection ensures the plan update is practical and progressive.

## 2 Related Works

**Finetuned LLM Agent** Reinforcement Learning has been widely used to train LLMs to master interactive tasks. ChatGPT (OpenAI, 2023) applies Reinforcement with Human Feedback (RLHF) to finetune a pre-trained LLM, enabling it to communicate interactively with humans. GLAM (Carta et al., 2023) uses LLM as a policy and finetunes it with online RL to improve its abilities to solve text-

| Method | In-Context Demonstration | Feedback Utilization | Plan Applicability | Need Test-Time Refinement |
|---|---|---|---|---|
| ReAct (Yao et al., 2023) | Yes | Action | N/A | No |
| Code as Policies (Liang et al., 2022) | Yes | Action | N/A | No |
| Reflexion (Shinn et al., 2023) | Yes | Action | N/A | Yes |
| Inner Monologue (Huang et al., 2023) | Yes | Action | N/A | Yes |
| RCI (Kim et al., 2023) | Yes | Action & Plan Opt | Single task instance | Yes |
| DEPS (Wang et al., 2023) | Yes | Action & Plan Opt | Single task instance | Yes |
| AdaPlanner (Sun et al., 2023) | Yes | Action & Plan Opt | Single task instance | Yes |
| AutoPlan | No | Action & Plan Opt | All task instances | No |

Table 1: Comparison of various prompt-based methods that employ a LLM agent to solve decision-making tasks. AutoPlan is the only method that optimizes a plan applicable to all task instances without any demonstration and requires no test-time refinement of the decision-making process.

based decision-making tasks. Experiments demonstrate that LLM policy significantly boosts sample efficiency. Other than RL, Xiang et al. also fine-tunes the LLM in a supervised manner with experiences collected through Monte Carlo Tree Search (MCTS). However, RL and supervised learning require calculating the gradients and updating the model's parameters, which is especially costly for LLMs.

**LLM with In-Context Learning**  As the size of the model and corpus scales, LLM demonstrates In-Context Learning (ICL) abilities, i.e., LLM directly learns from a few demonstrations of a task given in the context. Brown et al. shows that a pre-trained LLM performs strongly on traditional NLP tasks, including question answering and cloze tasks, with ICL. More recent works focus on the design of demonstrations (Sorensen et al., 2022; Lu et al., 2022). Sorensen et al. proposes to retrieve demonstrations with higher mutual information between model input and output. GlobalE&LocalE (Lu et al., 2022) uses entropy statistics to find the most performant permutation of demonstrations. Nonetheless, the ICL LLM agent is still sensitive to the provided demonstrations and requires additional human effort.

**Prompt-based LLM Agent**  Techniques have recently been developed to adapt LLMs to solve decision-making tasks through prompts. Table 1 illustrates the main difference between works along this line. ReAct (Yao et al., 2023) explicitly reasons over past interactions and determines the following action based on previous thoughts, actions, and observations. Reflexion (Shinn et al., 2023) built on top of ReAct and refines the interaction by iteratively reflecting on its past failed trials of a task

instance. However, Reflexion conducts test-time reflection and the reflection for one environment does not transfer to others. RCI (Kim et al., 2023), DEPS (Wang et al., 2023) and AdaPlanner (Sun et al., 2023) start with an initial plan of the task and refine the plan and the decision-making process for each specific task instance. Our AutoPlan instead optimizes a task-level plan and directly applies it to all task instances without further test-time refinement, which could be more efficient during inference.

## 3  AutoPlan

In this section, we describe AutoPlan in detail. We first describe the general procedure of using LLM to solve an interactive decision-making task. Then we present AutoPlan that solves the task by a text-based plan, obtained by an iterative three-stage process: AutoPlan 1) collects interaction experiences using the task plan at the current step, 2) reflects on the collected experiences, and 3) updates the plan.

### 3.1  Problem Formulation

We aim to design an LLM-based agent to accomplish an interactive task described in natural language. The agent is provided with a natural language description of the task, possible actions, and environmental observations. The task description $P$ includes a generic abstract description and a concrete task instance with an objective. Let $\mathcal{M}$ be the LLM agent, $\mathcal{A}$ be the set of possible actions, and $\mathcal{O}$ be the set of possible observations from the environment. One could augment the input with a custom prompt $\mathcal{X}$. At each step $t$, the agent $\mathcal{M}$ generates a text action $a_t \in \mathcal{A}$ and receives a text observation $o_t \in \mathcal{O}$ from the environment. $o_0$ denotes the initial observation, which could be empty. We define

| Prompt Name | Prompt Content |
|---|---|
| Thought-prompt | Identify which step of plan you are at. Show your thought about the one next action. Your thought should be faithful the plan step. |
| Summary-prompt | Summarize the interaction history in steps. |
| Flaw-prompt | Identify all flawed parts of the plan/action. Remember in this game, things are not like real world. The system message in observation is always correct and the plan plan/action may have flaws. |
| Rev-prompt | Suggest revision to the current flawed part of the plan. Only the flawed part. |
| Upd-prompt | Based on the above experiences of the game, rewrite the current game plan. Pay attention to summary of successful jobs, and flawed actions and suggested revision of all jobs. The plan should be generalizable to all job objectives. The actions in the plan should also be in the form as in game description. |

Table 2: Prompts that AutoPlan uses in ALFWorld environment.

a reward function $\mathcal{R}(o_{0:t}) = 1$ if the objective is achieved based on the observations. The problem of AutoPlan is to design an optimal prompt $\mathcal{X}$ to maximize the expected rewards over all possible task instances,

$$\mathcal{X}^* = \arg\max_{\mathcal{X}} \mathbb{E}_P \left[ \mathcal{R}(o_{0:T}) \right], \qquad (1)$$

where $T$ is the maximum number of interaction steps allowed.

Ideally, the optimal $\mathcal{X}^*$ should be adequate for all task instances of the same problem. Since the space of a custom prompt is vast, we frame such a prompt as a plan, which describes a sequence of actions in natural languages. Figure 2 shows a heating task in ALFWorld (Shridhar et al., 2021) and how the LLM agent solves this. Task description includes the available actions and an instance-wise objective (e.g., put a hot bowl in the sidetable). We aim to find an optimal plan as the custom prompt. After the task starts, the agent's current and visible locations constitute the first observation $o_0$. Then, the agent acts and observes the environment iteratively until it reaches the maximum number of interaction steps $T$.

Following prior work ReAct (Yao et al., 2023), we extend the original action space $\mathcal{A}$ to include $\mathcal{L}$, the space of thoughts expressed in language. As shown in Figure 2, a "thought" action (in the form of "Think[...]") does not elicit any environmental feedback and solely manifests the reasoning process of the LLM.

## 3.2 AutoPlan

AutoPlan treats a custom prompt $\mathcal{X}$ in the form of a task-solving plan that includes a sequence of abstract actions to execute in different scenarios. Such a plan described in natural language resembles the policy network in deep reinforcement learning, but it is more explainable due to its textual form. It is also more token-efficient than in-context demonstrations. Furthermore, state-of-the-art instruction-tuned LLMs demonstrate a strong ability to follow a given plan.

As shown in Figure 2, we design a three-stage process to optimize plan $\mathcal{X}$ iteratively: 1) experience collection with the current plan, 2) reflection on the collected experiences, and 3) plan update based on reflection.

### 3.2.1 Experience Collection

AutoPlan starts with an empty plan $\mathcal{X}_0$. At each iteration $i$, a batch of $B$ task instances is randomly selected, denoted as $P_1, P_2, \cdots, P_B$. For each task instance $P_j$, the LLM agent generates a sequence of thoughts and actions in response to observations from the environment.

Let $H_{t-1}^j = P_j \oplus \mathcal{X}_i \oplus (o_0, \tilde{a}_0, a_0, o_1, \cdots, o_{t-1})$ be the past interactions before step $t$. Since we augment the action space with thoughts that do not affect on the environment, at each step $t$, AutoPlan first obtains the thought,

$$\tilde{a}_t \sim \mathcal{M}(H_{t-1}^j \oplus \text{Thought-prompt}) \qquad (2)$$

where Thought-prompt is provided in Table 2 to make LLM agent act faithfully to the plan $\mathcal{X}_i$. Then

we sample the next action given the thought $\tilde{a}_t$,

$$a'_t \sim \mathcal{M}(H^j_{t-1} \oplus \tilde{a}_t \oplus \text{"Action:"}) \quad (3)$$

$$a_t = \mathcal{F}(a'_t) \quad (4)$$

$$H^j_t = H^j_{t-1} \oplus \tilde{a}_t \oplus a_t \oplus o_t. \quad (5)$$

where $o_t$ is the observation after action $a_t$ and $\mathcal{F}$ is a formalizer used to reformat the action to be acceptable by the environment. Details of the formalizer can be found in Appendix A.1.

As shown in Figure 2, $\tilde{a}_t$, $a_t$ and $o_t$ correspond to "Think[...]", "Action[...]" and "Observation[...]" in the experience of a task instance, where the LLM agent successfully found the bowl on the sidetable but failed to heat it with the toaster.

### 3.2.2 SIR Reflection

Given the experience $H^j_T$ and the corresponding reward $\mathcal{R}(o_{0:T})$ (denoted as $\mathcal{R}^j$), we instruct the LLM agent to reflect on the interaction history through a SIR reflection procedure: 1) Summarize the interaction history, 2) Identify the flawed steps of the plan, 3) Revise the flawed steps,

$$s_j = \mathcal{M}(H^j \oplus \mathcal{R}^j \oplus \text{Summary-prompt}) \quad (6)$$

$$f_j = \mathcal{M}(H^j \oplus \mathcal{R}^j \oplus \text{Flaw-prompt}) \quad (7)$$

$$r_j = \mathcal{M}(H^j \oplus \mathcal{R}^j \oplus \text{Flaw-prompt} \\ \oplus \text{Rev-prompt}) \quad (8)$$

where Summary/Flaw/Rev-prompts are shown in Table 2. The summarization, flaw, and revision provide necessary and clear information for the plan updater to modify the current plan.

As shown in Figure 2, the reflection summarizes the key actions, successfully identifies the flaw part of the plan, where $\mathcal{X}_i$ treats the toaster as the appropriate heating appliance, and suggests a revision to use the microwave instead.

### 3.2.3 Plan Update

With the task descriptions $P_1, P_2, \cdots, P_B$, the current task plan $\mathcal{X}_i$, and the summarizations $s_1, \cdots, s_B$, identified flaws $f_1, \cdots, f_B$ and revisions $r_1, \cdots, r_B$, we utilize the LLM to revise $\mathcal{X}_i$ and obtain an improved plan $\mathcal{X}_{i+1}$,

$$\mathcal{X}_{i+1} = \mathcal{M}(\mathcal{X}_i \oplus (P_1, s_1, f_1, r_1) \oplus \cdots \\ \oplus (P_B, s_B, f_B, r_B) \oplus \text{Upd-prompt}) \quad (9)$$

where Upd-prompt (as shown in Table 2) asks the LLM to generate an updated plan given the task instances and reflections.

In the example of Figure 2, the plan updater aggregates the task instances with their reflections and rewrites the new plan to use the microwave to heat the target objects instead.

After obtaining a revised plan $\mathcal{X}_{i+1}$, we continue the iterative process until we reach maximum optimization iterations $I$. During inference, we follow the same procedure as experience collection except that now we use the final optimized plan $\mathcal{X}_I$.

To summarize, AutoPlan uses LLM to solve a text-based interactive decision-making task through a task plan described in natural language. The plan is optimized iteratively through a three-stage process. The final plan is then used during inference time.

## 4 Experiment

We aim to answer the following questions:
1) Does AutoPlan improve upon baselines?
2) Is AutoPlan efficient during inference?
3) Does batching stabilize the optimization?
4) Does trio reflection ensure steady progression?

### 4.1 Data

**ALFWorld** is a text-based game enabling agents to navigate and interact with a simulated household to accomplish six types of tasks. Each task instance comes with a high-level objective (e.g., put a hot tomato on the desk), and the agent can achieve the objective through low-level actions described in text (e.g., heat tomato 1 with microwave 2, go to desk 1). Since the environment feedback of invalid actions provided in the original ALFWorld is too primitive, we manually augment the feedback (Table 6) to include possible causes of the invalidity. Further details of ALFWorld can be found in the Appendix B.1.

We randomly sample 24 task instances for each type of task from the training games to optimize the task-specific plan and, following prior works (Shridhar et al., 2021; Yao et al., 2023), use 134 unseen validation games[1] to evaluate our method. ALFWorld evaluates the success/failure of a task instance by checking if the agent is in the goal state (e.g. if the hot mug is already on the desk).

**HotpotQA** is a multi-hop question answering benchmark requiring reasoning over two or more

---
[1] Unseen games have the same task types but different objects, receptacles and household layout.

| Method | Success Rate | | | | | |
|---|---|---|---|---|---|---|
| | *Pick* | *Light* | *Clean* | *Heat* | *Cool* | *Pick Two* |
| *Supervised method* | | | | | | |
| BUTLER | 46 | 22 | 39 | 74 | **100** | 24 |
| *Prompt methods w/ ground-truth demonstrations* | | | | | | |
| AdaPlanner (1 Shot) † | **100** | **100** | 97 | **96** | **100** | 47 |
| ReAct (2 Shot) | **100** | **100** | **100** | 91 | 96 | 76 |
| Reflexion (2 Shot) | **100** | **100** | **100** | 91 | **100** | 94 |
| *Prompt methods w/o ground-truth demonstrations* | | | | | | |
| AdaPlanner (0 Shot) | 0 | 0 | 0 | 0 | 0 | 0 |
| ReAct (0 Shot) | 92 | 94 | 87 | 35 | 71 | 59 |
| Reflexion (0 Shot) | 96 | **100** | 97 | 52 | 81 | 88 |
| AutoPlan | **100** | **100** | 97 | **96** | 90 | 82 |

(a) ALFWorld

| Method | Acc |
|---|---|
| *Supervised Method* | |
| Chain-of-Skills | **90** |
| *Prompt Methods* | |
| ReAct (0 Shot) | 70 |
| ReAct (6 Shot) | 75 |
| AutoPlan | 83 |

(b) HotpotQA

Table 3: Accuracy and Success rate (%) of AutoPlan and baselines on HotpotQA and ALFWorld, respectively. AutoPlan consistently outperforms the 0-shot baseline, achieves on-par success rates with baselines leveraging ground-truth demonstrations on ALFWorld, and even beats the 2-shot ICL baseline on HotpotQA by 8%. † Results of AdaPlanner are from the original paper since the author does not provide enough details for reproduction.

Wikipedia pages. As in (Yao et al., 2023), the LLM agent is required to answer questions by interacting with a Wikipedia API. The API supports three types of actions: (1) search[entity]: returns the first five sentences from the Wikipedia page of the entity if it exists or suggests top-5 similar entities[2]. (2) lookup[string]: returns the following sentence containing string. (3) finish[answer]: finishes the task with an answer.

We randomly sample 50 hard (question, answer, supporting facts) triples from the official training set to optimize the plan and sample 200 questions from the official development set as the test set[3]. The final answer is evaluated by three external human annotators rather than exact-match (EM) since the answer provided by the agent and the gold answer can differ drastically in form but share the same meaning. We include the complete annotation instruction in the Appendix B.2 and take the majority vote of 3 annotators as the final evaluation result. The agreement rate (all three annotators agree with each other) is above 90% for all considered models.

---

[2]We notice that this API retrieves the latest information instead of the Wikipedia dump (2017-10-01) used to build HotpotQA dataset, so we modify it to return the historical page of entities before 2017-10-01.

[3]200 is a trade-off between our budget and evaluation uncertainty.

### 4.2 Method Configurations

We use GPT-4-0314 (OpenAI, 2023) as the LLM across all experiments. The maximum number of actions is 10 for HotpotQA and 35 for ALFWorld. The default batch size of task instances is 4 for both HotpotQA and ALFWorld. We use nucleus sampling with $p = 0.9$ during optimization and greedy decoding during evaluation. The full prompt templates of both environments can be found in the Appendix A.2.

### 4.3 Baselines

We compare with the following baselines.
- *ReAct (K Shot)*: The custom prompt $\mathcal{X}$ consists of $K$ demonstrations manually written by human. We reuse the demonstrations provided in (Yao et al., 2023). We have $K = 6$ for HotpotQA and $K = 2$ for ALFWorld.
- *Reflexion (K Shot)*: Built on top of ReAct, Reflexion conducts iterative test-time reflection for each environment, using the interaction history to revise the actions in the following iterations. We set the number of iterations to be five and use the same custom prompt as in ReAct.
- *AdaPlanner (Sun et al., 2023) (K Shot)*: AdaPlanner also proposes to optimize the plan with LLM but using a code-style custom prompt, which is more rigorous but also more restric-

tive than AutoPlan. Note that AdaPlanner still requires human-written demonstrations to initialize the plan.

- *Supervised Baseline*: For HotpotQA, we select the best available supervised method Chain-of-Skills (Ma et al., 2023) from the leaderboard of fullwiki setting. For ALFWorld, we choose BUT-LER (Shridhar et al., 2021), an imitation learning agent trained with $10^5$ human demonstrations for each task type.

## 4.4 Main Results

**Success Rates** The success rate and accuracy of AutoPlan and baselines in ALFWorld and HotpotQA are shown in Table 3 respectively. In ALFWorld, AutoPlan achieves on-par success rates with ReAct (2 Shot), AdaPlanner (1 Shot), and Reflexion (2 Shot) on all six types of tasks and outperforms zero-shot baselines by at most 44% on *Heat* task. Notably, AutoPlan accomplishes the first four tasks nearly perfectly with success rates approaching 100% and success rates above 90% and 80% for the latter two. In HotpotQA, AutoPlan answers questions even 8% more accurately than ReAct (6 Shot) with human-written demonstrations of how to use the search tool, thanks to the optimized plan.

**Error Analysis** Of 137 ALFWorld test instances, AutoPlan fails seven due to the inability to locate the target object. One failure stems from a lexical misunderstanding where the LLM confuses a "cup" with a "mug". Another results from an atypical object location, with the apple to be heated found in a garbage can. The remaining five failures occur due to the LLM's erroneous prior assumptions about potential object locations, even though the plan points the model towards the most probable ones. Once the agent locates the task instance's target object(s), it performs all subsequent actions correctly. We observe similar failure patterns in cases of ReAct (2 Shot). With neither the optimized plan nor in-context demonstrations, ReAct (0 Shot) struggles to find the correct action sequence to clean/cool/heat the object even if it finds the target object(s).

In HotpotQA, AutoPlan achieves better logical consistency than ReAct (0/6 Shot) thanks to the step-by-step plan. ReAct (6 Shot) performs well when only a few actions are needed but can diverge to unreasonable thought and action processes when the number of actions is considerable. One primary reason is that the demonstrations used in ReAct (6 Shot) involve no more than five actions, which

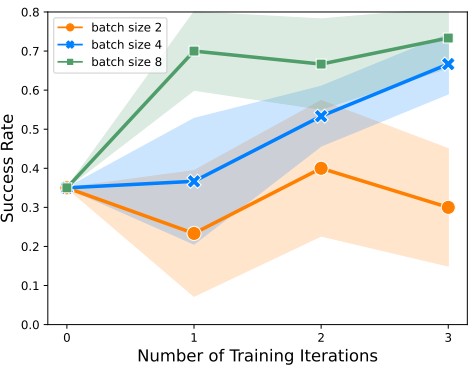

(a) Batch size

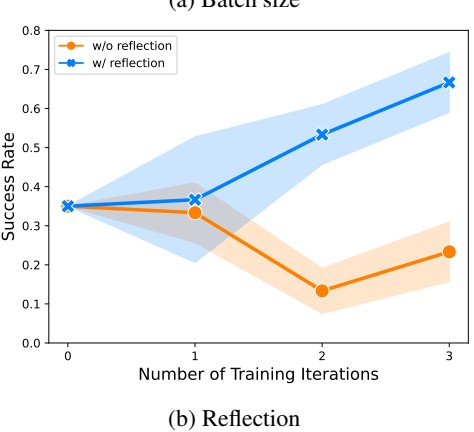

(b) Reflection

Figure 3: The success rate of AutoPlan on task *Heat* of ALFWorld optimized (a) with different batch sizes and (b) with/without complete reflection process. We plot the mean (marked line) and standard deviation (band) of five independent runs. A larger batch size significantly improves the success rate on average and reduces the variance. The reflection process in AutoPlan ensures the steady improvement over iterations.

again shows that the ICL method is sensitive to the quality of demonstrations.

**Training and Inference Cost** We measure the training and inference cost of AutoPlan and baselines per instance in Table 4. The cost is calculated based on the official documentation[4]. AutoPlan requires only marginal additional cost compared to ReAct (0 Shot) while achieving the best result on ALFWorld and HotpotQA.

## 4.5 Ablation Study

The plan optimization process of AutoPlan can be precarious due to sampling-based decoding. To tackle this, AutoPlan batches multiple task instances together in one iteration to stabilize the optimization and applies an explicit 3-step reflection to elicit helpful information from the interaction

---

[4]https://openai.com/pricing

| Method | Training | Inference |
|--------|----------|-----------|
| ReAct (2 Shot) | N/A | 3 |
| Reflexion (2 Shot) | N/A | 17 |
| AdaPlanner (1 Shot) | N/A | 2.1 |
| AutoPlan | 1.8 | 1.6 |

(a) ALFWorld

| Method | Training | Inference |
|--------|----------|-----------|
| ReAct (0 Shot) | N/A | 0.15 |
| ReAct (6 Shot) | N/A | 0.46 |
| AutoPlan | 0.26 | 0.23 |

(b) HotpotQA

Table 4: Average cost (unit: US Dollar) per question used by methods in ALFWorld and HotpotQA environments. Cost is calculated based on the OpenAI pricing document.

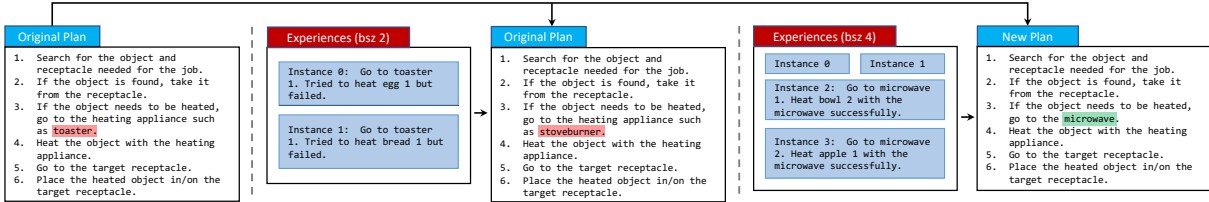

Figure 4: An illustration of the impact of batch size on the plan update. The agent with batch size two only tried the toaster to heat the object, but with batch size four, the agent also tried the microwave, the only allowed heating appliance in this task—the larger the batch size, the more chance the agent can find the correct action sequence.

history. Here, we demonstrate the effectiveness of batching and reflection on task *Heat* of ALF-World as this is the task that AutoPlan achieves the largest improvement against the baseline ReAct (0 Shot) with no plan. We first run AutoPlan five times with both batch sizes 2, 4, and 8, and then run five times with and without the last two steps of reflection (flaw identification and revision)[5]. Then, we measure the mean and standard deviation of test success rates of plans produced in the first three iterations.

**Larger batch size significantly stabilizes the optimization process.** As shown in Figure 3a, a larger batch size improves the average success rate and reduces the standard deviation during optimization. We also conducted a t-test comparing batch size 2 and 8 results, and the p-value is no more than 0.110 for all iterations (see Table 5). Carefully examining the interaction histories, we find that with a larger batch size, the agent is more likely to hit the right action during the experience collection stage. As illustrated in Figure 4, the agent with batch size 2 only tried the toaster to heat the object, but with batch size 4, the agent also tried the microwave, the only correct heating appliance for this task.

**Reflection ensures the optimization goes in the right direction.** As shown in Figure 3b, Auto-Plan with the complete reflection obtains steady improvements after each iteration, while the success rate of AutoPlan with only the interaction summary bounces back and forth between 0% and 30%, even below the success rate of ReAct (0 Shot). Again we can visualize such a difference in Figure 5. The agent went to the microwave and tried to heat the object but failed because of the wrong action sequence (the correct action sequence can be found in Table 8). AutoPlan with complete reflection explicitly identifies such flawed behavior from the observation and proposes a revision, which is later integrated into the new plan. However, AutoPlan without flaw identification and revision does not realize the valid reason for failure and leads to undesired plan updates.

## 5 Conclusion

We propose AutoPlan, a prompt-based method, to enable LLM to solve interactive decision-making tasks without gradient computation or in-context demonstrations. AutoPlan conditions LLM on an additional task plan described in natural language, which is obtained through an iterative three-stage process. Experiments show that AutoPlan achieves better results than baselines and is also efficient during inference. The ablation study further confirms the effectiveness of batching and explicit reflection

[5]We keep the summary step of reflection since the plan update is meaningless without the interaction summary.

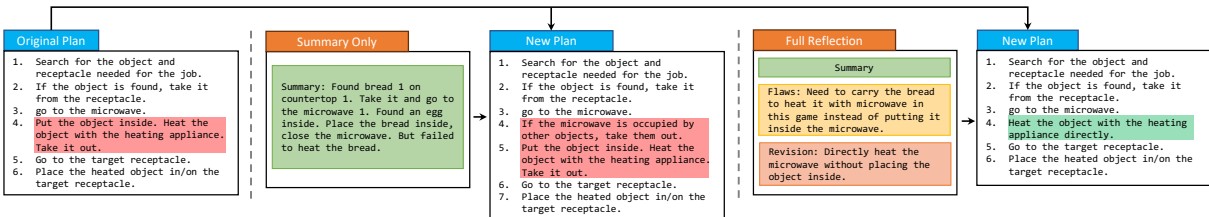

Figure 5: An illustration of the impact of reflection on the plan update. With only a summary of interactions, the plan updater needs to guess the cause of failure. Eventually it leads to the wrong thought that the objects inside the microwave need to move before heating. With flaw identification and suggested revision, the plan updater understands the flawed part of the plan and rewrites the plan to heat the object directly.

in stabilizing the plan optimization process.

## Limitation

The improvements of AutoPlan mainly come from two sources: 1) the correct action sequence sampled during exploration; 2) the environment feedback when incorrect actions are sampled by the LLM agent. As shown in Table 6, the feedback directly tells the agent which aspect of the action is invalid. Without such fine-grained feedback, the agent needs to collect more experience, i.e., larger batch size, to make sure the correct action sequence is sampled with high probability.

Another limitation is that in order to make AutoPlan works without any demonstration, we rely on GPT-4-0314 to generate action sequences, reflect on the interactions, and update the plan. We tried to use GPT-3.5-turbo-0301, but find out 1) it fails to follow the plan faithfully even explicitly prompted to do so; 2) it generates too many hallucinated contents about the environment, which could (possibly) be handled by better prompt design, but that requires excessive human effort, contradicting the goal of AutoPlan to reduce human effort as much as possible. It is worth trying other state-of-the-art LLMs such as Claude[6] to see which one also works.

## Ethics Statement

While AutoPlan is capable of functioning solely with task descriptions and observations, it is imperative to exercise caution while using it in high-stakes circumstances, given the inherent unpredictability of LLMs. Furthermore, we earnestly recommend that users carefully assess if the objectives could inadvertently cause harm to others before putting AutoPlan into action.

---

[6]https://www.anthropic.com/index/introducing-claude

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

# A Detailed Implementation of AutoPlan

## A.1 Formalizer

The formalizer is again a LLM call with specially designed prompt as shown in Figure 6.

## A.2 Full Prompt of AutoPlan

Full prompts of ALFWorld and HotpotQA are shown in Figure 7 (experience collection and reflection) and Figure 8 (plan update).

|  | Iter 1 | Iter 2 | Iter 3 |
|---|---|---|---|
| p-value (2 & 4) | 0.44 | 0.35 | 0.013 |
| p-value (2 & 8) | 0.007 | 0.110 | 0.005 |

Table 5: P-values of t-test between results of batch size 2 & 4 and 2 & 8. Batch size 8 delivers significantly higher success rates than batch size 2.

### A.3 Feedback

The examples of augmented feedback of ALF-World are shown in Table 6. We do not add additional feedback for HotpotQA upon the original one designed in ReAct (Yao et al., 2023).

## B Additional Details in Experiments

### B.1 Environments

The task types and templates of task objectives of ALFWorld are listed in Table 7. The allowed actions can be found in Figure 7. The correct action sequences for each task can be found in Table 8.

### B.2 Human Evaluation

We invite three external human annotators to conduct human evaluation on HotpotQA. Instructions for human annotators are shown in Figure 9. We take the majority votes from human annotators as accuracy and also compute the agreement among three annotators.

### B.3 Significant Test

We conduct t-test between success rates of plans generated by batch size 2, 4 and 8 at each iteration. The p-values are shown in Table 5.

```
Valid action formats are as follows:
go to "recep"
take "object" from "recep"
put "object" in/on "recep"
open "recep"
close "recep"
use "recep"
clean "object" with "recep"
heat "object" with "recep"
cool "object" with "recep"

The "object" and "recep" should be replaced with real
names and indices, e.g., "apple 1" and "desk 1".

Formalize the following action strictly into the
above valid action formats. If there are multiple
actions, formalize the first one.

Action to formalize: {raw_action}
Formalized action: {formalized_action}
```

(a) ALFWorld

```
Valid action formats are as follows:
search[entity]
lookup[keyword]
finish[answer]

Formalize the following action strictly into the
above valid action formats. If there are multiple
actions, formalize the first one.

Action to formalize: {raw_action}
Formalized action: {formalized_action}
```

(b) HotpotQA

Figure 6: The prompts of formalizer for (a) ALFWorld and (b) HotpotQA. With input raw action, the LLM generates the formalized action.

| Error Type | Example Action | Augmented Feedback |
|---|---|---|
| Missing Index | take tomato from countertop 1 | You miss the index of tomato, e.g., tomato 1. |
| Wrong Location | take tomato 1 from countertop 1 | You are not at countertop 1. |
| Invalid Receptacle | take tomato 1 from countertop 1 | countertop 1 is not a valid action in this household. |
| Closed Receptacle | take tomato 1 from cabinet 1 | cabinet 1 is closed. |
| Inventory Limit | take tomato 1 from cabinet 1 | You cannot hold more than one object. |
| Not In Inventory | put tomato 1 in/on cabinet 1 | You are not carrying tomato 1. |
| Not In Inventory | put tomato 1 in/on cabinet 1 | You are not carrying tomato 1. |
| Invalid Heating Appliance | heat tomato 1 with toaster 1 | toaster cannot be used for heating. |

Table 6: Examples of fine-grained feedback with respect to various erroneous actions.

| Task Type | Templates |
|---|---|
| Pick | put a obj in recep.
put some obj on recep. |
| Light | look at obj under the desklamp.
examine the obj with the desklamp. |
| Clean | put a clean obj in recep.
clean some obj and put it in recep. |
| Heat | put a hot obj in recep.
heat some obj and put it in recep. |
| Cool | put a cool obj in recep.
cool some obj and put it in recep. |
| Pick Two | put two obj in recep.
find two obj and put them in recep. |

Table 7: Six task types of ALFWorld and their objective templates.

**ALFWorld: Experience Collection and Reflection**

```
Game Description:
You need to interact with a simulated household to solve a
job. The simulated house has many objects and receptacles.
Valid Actions on the objects and receptacles are as follows:
(1) go to recep
(2) take object from recep: take object from some receptacle
and carry it with you
(3) put object in/on recep: put the object you are carrying
in/on some receptacle
(4) open recep: open some closed receptacle
(5) close recep: close some open receptacle
(6) use recep: use some receptacle
(7) clean object with recep
(8) heat object with recep
(9) cool object with recep
You job is to heat some object and put it in/on some
receptacle.

Game Objective: You need to put a hot potato in fridge.

Game Plan:
1. Go to the most likely location of the object
2. Open the receptacle if necessary
3. If the object is not found, continue searching other
   locations
4. If the object is found, take the object from the
   receptacle
5. Go to the toaster
6. Heat the object with the toaster
7. Go to the target receptacle
8. Put the object in/on the receptacle

Game Starts. You are in the middle of a room. Looking around
you, you see a countertop 1, microwave 1, toaster 1,
sidetable 1, and a sidetable 2.

Identify which step of plan you are at. Show your thought
about the one next action. Your thought should be faithful to
the plan step.
> I am currently at step 1. I should check sidetable for the
  bowl.
Action:
> go to sidetable 1
Observation: On the countertop 1, you see a bowl 1.
Identify which step of plan you are at. Show your thought
about the one next action. Your thought should be faithful to
the plan step.
> I am currently as step 2. I need to take the bowl from the
  sidetable.
Action:
> take bowl 1 from sidetable 1.
Observation: You pick up bowl 1.
```

```
Identify which step of plan you are at. Show your thought
about the one next action. Your thought should be faithful to
the plan step.
> I am currently at step 3. I should go to the toaster.
Action:
> go to toaster 1
Observation: On the toaster 1, you see nothing.

Identify which step of plan you are at. Show your thought
about the one next action. Your thought should be faithful to
the plan step.
> I am currently at step 4. I need to heat the bowl with the
  toaster.
Action:
> heat bowl 1 with toaster 1
Observation: Action failed. Toaster cannot heat bowl.

Identify which step of plan you are at. Show your thought
about the one next action. Your thought should be faithful to
the plan step.
> It seems toaster is not working. I need to check the
  microwave instead.
Action:
> go to microwave 1
Observation: The microwave 1 is open. In it, you see nothing.

Identify which step of plan you are at. Show your thought
about the one next action. Your thought should be faithful to
the plan step.
> I'll try using microwave to heat the object.
Action:
> heat bowl 1 with microwave 1
Observation: You heat bowl 1 with microwave 1.

Maximum number of actions reached. Task fails.
Summarize the interaction history in steps.
> I found the bowl 1 on sidetable 1. I tried to heat it with
  the toaster 1 but failed. I finally heat it with microwave
  1 but failed the task by exceeding the maximum allowed
  number of actions.

Identify the flawed part of the plan/action. Remember in this
game things are not like real world. The system message is
always correct and the game plan/action may have flaws.
> As the observation said, I need to heat the bowl with
  microwave instead of toaster in this task.

Suggest revision to the current flawed part of the plan. Only
the flawed part.
> change "toaster" in step 5-6 into "microwave"
```

**HotpotQA: Experience Collection and Reflection**

```
Task Description:
Solve a question answering task with interleaving Thought,
Action, Observation steps. Thought can reason about the
current situation, and Action can be of three types:
(1) search[entity], which searches the exact entity on
Wikipedia and returns the first paragraph if it exists. If
not, it will return some similar entities to search.
(2) lookup[keyword], which returns the next sentence
containing keyword in the current passage.
(3) finish[answer], which returns the answer and finishes
the task. Call finish[] if the answer is not found.

Question: Maurice Hines and his brother were famous for
what?

Task Plan:
1. Identify the main keywords of entities.
2. Search for the main entity of keyword on Wikipedia using
search[entity].
3. Look for the next sentence containing the keyword in the
current Wikipedia page.
4. Repeat step 2 and 3 as necessary until the answer is
found.
5. Finish the task with finish[answer].

Identify which step of plan you are at. Show your thought
about the one next action. Your thought should be faithful
to the plan step.
> I am currently at step 1. The entity is "Maurice Hines".
  According to step 2, I need to search for the entity
  using search[entity].
Action:
> search[Maurice Hines]
Observation: Maurice Hines (born December 13, 1943) is an
American actor, director, singer, and choreographer.[1]. He
is the brother of dancer Gregory Hines. ... Hines began his
career at the age of five, studying tap dance at the Henry
LeTang Dance Studio in Manhattan.[3]. ... Shortly after,
the brothers began touring as the opening act for such
headliners as Lionel Hampton and Gypsy Rose Lee.

Identify which step of plan you are at. Show your thought
about the one next action. Your thought should be faithful
to the plan step.
```

```
> I am currently at step 3. I need to look for the keyword
  "tap dancer" as shown in the return of search query.
Action:
> lookup[tap dance]
Observation: (Result 1 / 1) Hines was born in 1943 in New
York City, the son of Alma Iola (Lawless) and Maurice
Robert Hines, a dancer, musician, and actor.[2] Hines began
his career at the age of five, studying tap dance at the
Henry LeTang Dance Studio in Manhattan.[3] ... Maurice made
his Broadway debut in The Girl in Pink Tights in 1954.[4].

Identify which step of plan you are at. Show your thought
about the one next action. Your thought should be faithful
to the plan step.
> I am now at step 5 since the answer "tap dancing" is
  found.
Action:
> finish[Maurice Hines and his brother were famous for tap
  dancing.]

Task finished. The ground truth answer is "dancer Gregory
Hines" and the correct entities to search are "Hot Feet"
and "Maurice Hines". Summarize the interaction history
concisely.
> I searched for "Maurice Hines" on Wikipedia and found
  information about him and his brother Gregory Hines. I
  then looked up the keyword "tap dance" and found
  both Maurice and Gregory Hines were famous for tap
  dancing. The answer provided was that Maurice Hines and
  his brother Gregory Hines were famous for tap dancing.

Identify all flawed parts of the plan (not flawed action).
> Step 3 does not account for the possibility that the
  answer might already be present in the first paragraph
  returned by the search[] action. In such cases, the
  lookup[] action might not be necessary, and the assistant
  can directly proceed to step 5 to provide the answer.

Suggest revision to the current flawed part of the plan.
Only the flawed part.
> Step 3. If the answer is not found in the first paragraph
  returned by search[entity], use lookup[keyword] to look
  for the next sentence containing the keyword in the
  current Wikipedia page opened by search[entity].
```

Figure 7: Full prompts of experience collection and reflection of ALFWorld and HotpotQA. LLM outputs are preceded with '>'.

```
Game Description:
You need to interact with a simulated household to solve a job. The simulated house
has many objects and receptacles. Valid Actions on the objects and receptacles are
as follows:
(1) go to recep
(2) take object from recep: take object from some receptacle and carry it with you
(3) put object in/on recep: put the object you are carrying in/on some receptacle
(4) open recep: open some closed receptacle
(5) close recep: close some open receptacle
(6) use recep: use some receptacle
(7) clean object with recep
(8) heat object with recep
(9) cool object with recep
You job is to heat some object and put it in/on some receptacle.

Current Game Plan:
1. Go to the most likely location of the object
2. Open the receptacle if necessary
3. If the object is not found, continue searching other locations
4. If the object is found, take the object from the receptacle
5. Go to the toaster
6. Heat the object with the toaster
7. Go to the target receptacle
8. Put the object in/on the receptacle

Game Objective 1: You need to put a hot potato in fridge.
Summary 1: I found the bowl 1 on sidetable 1. I tried to heat it with the toaster 1
but failed. I finally heat it with microwave 1 but failed the task by exceeding the
maximum number of actions.
Flaws 1: As the observation said, I need to heat the bowl with microwave instead of
toaster in this task.
Revision 1: change "toaster" in step 5-6 into "microwave"

Based on the above experiences of the game, rewrite the current game plan. Pay more
attention to summary of successful jobs, and flawed actions and suggested revision
of failed jobs. The plan should not be specific to one game objective but
generalizable to all objectives. The actions in the plan should also be in the form
as in game description.

> New Game Plan:
1. Go to the most likely location of the object
2. Open the receptacle if necessary
3. If the object is not found, continue searching other locations
4. If the object is found, take the object from the receptacle
5. Go to the toaster
6. Heat the object with the toaster
7. Go to the target receptacle
8. Put the object in/on the receptacle
```

```
Task Description:
Solve a question answering task with interleaving Thought, Action, Observation steps.
Thought can reason about the current situation, and Action can be of three types:
(1) search[entity], which searches the exact entity on Wikipedia and returns the
first paragraph if it exists. If not, it will return some similar entities to search.
(2) lookup[keyword], which returns the next sentence containing keyword in the
current passage.
(3) finish[answer], which returns the answer and finishes the task. Call finish[] if
the answer is not found.

Current Task Plan:
1. Identify the main keywords of entities.
2. Search for the main entity of keyword on Wikipedia using search[entity].
3. Look for the next sentence containing the keyword in the current Wikipedia page.
4. Repeat step 2 and 3 as necessary until the answer is found.
5. Finish the task with finish[answer].

Question 1: Maurice Hines and his brother were famous for what?
Summary 1: I searched for "Maurice Hines" on Wikipedia and found information about
him and his brother Gregory Hines. I then looked up the keyword "tap dance" and found
that both Maurice and Gregory Hines were famous for tap dancing. The answer provided
was that Maurice Hines and his brother Gregory Hines were famous for tap dancing.
Flaws 1: Step 3 does not account for the possibility that the answer might already be
present in the first paragraph returned by the search[] action. In such cases, the
lookup[] action might not be necessary, and the assistant can directly proceed to
step 5 to provide the answer.
Revision 1: Step 3. If the answer is not found in the first paragraph returned by
search[entity], use lookup[keyword] to look for the next sentence containing the
keyword in the current Wikipedia page opened by search[entity].

Based on the above experiences of the task, rewrite the current task plan. Pay more
attention to summary of successful questions, and flawed actions and suggested
revision of failed questions. The plan should not be specific to one question but
generalizable to all questions. The actions in the plan should also be in the form as
in task description.

> New Task Plan:
1. Identify the main keywords of entities.
2. Search for the main entity of keyword on Wikipedia using search[entity].
3. If the answer is not found in the first paragraph returned by search[entity], Look
for the next sentence containing the keyword in the current Wikipedia page.
4. Repeat step 2 and 3 as necessary until the answer is found.
5. Finish the task with finish[answer].
```

Figure 8: Full prompts of plan update of ALFWorld and HotpotQA. LLM outputs are preceded with '>'.

| Task Type | Correct Action Sequence |
|---|---|
| Pick | go to the receptacle with target object; pick it up; go to the target receptacle; put it down. |
| Light | go to the receptacle with target object; pick it up; go to the receptacle with a desklamp; use the desklamp. |
| Clean | go to the receptacle with target object; pick it up; go to a sinkbasin; clean the object with the sinkbasin; go to the target receptacle; put it down. |
| Heat | go to the receptacle with target object; pick it up; go to a microwave; heat the object with the microwave; go to the target receptacle; put it down. |
| Cool | go to the receptacle with target object; pick it up; go to a fridge; cool the object with the fridge; go to the target receptacle; put it down. |
| Pick Two | go to the receptacle with the first target object; pick it up; go to the target receptacle; put it down; go to the receptacle with the second target object; pick it up; go to the target receptacle; put it down. |

Table 8: Correct action sequences for each type of task in ALFWorld.

# Annotation Instructions

## Objective

The primary objective is to evaluate the quality of predicted answers generated by an automated method against the ground-truth answers for a set of 200 data points from the HotpotQA dataset. Each data point consists of a question, its corresponding ground-truth answer, supporting facts, and a predicted answer.

## Workflow

1. **Review Data Point**: Examine the components of the data point (question, ground-truth answer, supporting facts, and predicted answer).
2. **Check Accuracy**: Determine whether the predicted answer correctly addresses the question, considering the ground-truth answer and supporting facts.
3. **Check Consistency**: Verify if the predicted answer is consistent with the supporting facts.
4. **Tagging**: Use the annotation tool to tag the predicted answer as either 'Correct' or 'Incorrect', and add comments for clarification, if necessary.

## Guidelines

### Review Data Point

- Thoroughly read all the components (question, ground-truth answer, supporting facts, and predicted answer) before making any evaluations.

### Check Accuracy

- The predicted answer should directly answer the question posed.
- Compare the predicted answer to the ground-truth answer. If they match or are synonymous, the predicted answer is 'Correct'.
- If the predicted answer is partially correct but missing vital information, mark it as 'Incorrect' and note what is missing in the comments.

### Check Consistency

- The predicted answer must align with the supporting facts provided. If the answer goes beyond or contradicts these facts, mark it as 'Incorrect'.
- Inconsistencies can include incorrect names, dates, events, or any information that deviates from the supporting facts.

### Tagging

- Use the provided tagging system in the annotation tool to categorize the predicted answer as 'Correct' or 'Incorrect'.
- If the predicted answer is incorrect, make use of the comment section to briefly clarify what specifically is incorrect about it (e.g., "The date is wrong," "The answer is incomplete," etc.)

## Examples

### Data Point Example

- **Question**: Who delivered the 'I Have a Dream' speech?
- **Ground-Truth Answer**: Martin Luther King Jr.
- **Supporting Facts**: In 1963, Martin Luther King Jr. delivered his famous 'I Have a Dream' speech in Washington D.C.
- **Predicted Answer**: Martin Luther King Jr.

### Correct Annotation

- Tagging: 'Correct'

Figure 9: Instruction for human annotators to conduct human evaluation on model predictions on HotpotQA.