# OpenReview forum: "AutoPlan: Automatic Planning of Interactive Decision-Making Tasks With Large Language Models"
_EMNLP/2023/Conference — EMNLP 2023 Findings_

### Official Review · Reviewer_kvA8 · 2023-08-01

**Soundness:** 3

**Excitement:**

3: Ambivalent: It has merits (e.g., it reports state-of-the-art results, the idea is nice), but there are key weaknesses (e.g., it describes incremental work), and it can significantly benefit from another round of revision. However, I won't object to accepting it if my co-reviewers champion it.

**Paper Topic And Main Contributions:**

The paper develops a demonstration-free method for generating and iteratively improving a plan for decision making tasks using an LLM. The method is evaluated in both the ALFWorld and HotpotQA domains and outperforms a zero-shot ReAct baseline. In addition to downstream task performance, the authors investigate the inference costs of their method and perform ablations to understand the effect of batch size on the method.

**Questions For The Authors:**

What are the key differences between reflection in AutoPlan and in Reflexion?

Why did you not directly compare to a version of Reflexion without in-context examples and test time refinement? This seems like a critical comparison that is needed.

How does the overall performance of AutoPlan compare to the original Reflexion results? What is the actual improvement (if any) is achieved by using in-context examples and/or test time refinement?

Response to author rebuttal:

Thank you for taking the time to answer the questions and perform new experiments to provide further insight and validate your method. Given the high performance of all evaluated methods in ALFWorld, the paper could benefit from validation in more domains to further distinguish AutoPlan and justify its high training cost.

In light of your responses which address some of the posed weaknesses, I increased the soundness score to a 3.

**Reasons To Accept:**

Provides a detailed qualitative analysis of failures in the ALFWorld domain to understand method performance characteristics.

Performs ablations to understand the effect of batch size on AutoPlan.

The paper evaluates the proposed method using GPT-4 thereby demonstrating results using a state-of-the-art LLM.

**Reasons To Reject:**

Prompts contain ungrammatical English that may or may not affect experimental results.

The paper does not evaluate against strong baselines like InnerMonologue, Reflexion, and other LLM planning methods which evaluate on ALFWorld and HotpotQA. These evaluations could be done fairly by modifying these methods to match the inputs to AutoPlan, for example by not providing in-context examples to these methods. AdaPlanner could also be evaluated in the zero-shot setting. The paper should compare to other methods which also allow for plan optimization before testing. A comparison to just React seems incomplete to establish the benefits of the approach compared to methods that use the same resources.

The paper would be improved by more clearly differentiating its contribution from existing methods in LLM planning. For example it’s not clear whether there’s a significant difference between its “reflection” component and similar mechanisms in existing works like Reflexion.

The proposed method does not perform significantly better than a 2-shot ReAct baseline in ALFWorld. This calls into question the potential impact of AutoPlan, as AutoPlan requires a complex plan optimization process versus ReAct which does not. Therefore the comparison of method costs only during inference is incomplete. The paper could be improved by comparing total costs including both plan optimization and inference.

The human evaluation of HotpotQA results is done by the authors of the paper, and the evaluation methodology and criteria are not detailed.

**Reproducibility:**

4: Could mostly reproduce the results, but there may be some variation because of sample variance or minor variations in their interpretation of the protocol or method.

**Reviewer Confidence:**

4: Quite sure. I tried to check the important points carefully. It's unlikely, though conceivable, that I missed something that should affect my ratings.

**Typos Grammar Style And Presentation Improvements:**

Grammatical errors distract from the paper’s findings. The paper should be proofread and edited by a fluent English speaker

---

> ### Author Rebuttal · Authors · 2023-08-29
>
> **Q1: Ungrammatical English in prompts.**
>
> A1:  We tune the prompt to guide the Language Learning Model (LLM) in generating the desired output. While the resulting prompt may be ungrammatical, it functions as expected.
>
> **Q2: Lack of strong baselines, e.g., zero-shot AdaPlanner and Reflexion.**
>
> A2: Thank you for the suggestion! We have conducted additional experiments with 0-shot AdaPlanner and 0/2-shot Reflexion on ALFWorld. The success rates over all 134 test environments are as follows:
>
> |  | AutoPlan | 0-Shot ReAct | 2-Shot ReAct | 0-Shot AdaPlanner | 0-Shot Reflexion | 2-Shot Reflexion |
> | --- | --- | --- | --- | --- | --- | --- |
> | Success Rate | 95% | 74% | 95% | 0% | 82% | 97% |
>
> AdaPlanner fails when no in-context examples are provided. 0-shot Reflexion improves upon 0-shot ReAct by 8%, and 2-shot Reflexion surpasses both 2-shot ReAct and AutoPlan by 2%.
>
> **Q3: Clarify the difference between AutoPlan and existing methods, e.g., Reflexion.**
>
> A3: Thank you for the suggestion! The key differences between AutoPlan and Reflexion are as follows:
>
> - Reflexion involves test-time reflection, requiring iterative reflection for each test environment. This makes it inefficient, and the reflections do not transfer to other test environments. In contrast, AutoPlan optimizes the plan using training data and generates action sequences for test environments in a single iteration.
> - While Reflexion uses in-context learning inside reflection, AutoPlan employs a three-stage reflection process that captures key information without the need for in-context examples.
>
> We will include these clarifications and comparisons with other methods in our paper.
>
> **Q4: AutoPlan performance versus few-Shot ReAct.**
>
> A4: AutoPlan achieves comparable results to few-shot ReAct in ALFWorld, but demonstrates a non-negligible advantage over few-shot ReAct in HotpotQA (see human evaluation results in A5). We have also included both training and inference costs (unit: US dollar) for ALFWorld in the table below:
>
> |  | AutoPlan | 2-Shot ReAct | AdaPlanner | 2-Shot Reflexion (5 trials) |
> | --- | --- | --- | --- | --- |
> | Inference cost per case | 1.6 | 3 | 2.1 | 17 |
> | Training cost (24 cases) | 43 | null | null | null |
>
> AutoPlan is the most efficient in terms of inference cost, with reasonable training overhead.
>
> **Q5: Human evaluation.**
>
> A5: We conducted additional human evaluations on HotpotQA with three external human annotators. The full annotation instructions will be included in the paper’s appendix. These instructions involve four phases: reviewing data points, checking the accuracy of predictions, ensuring consistency with supporting facts, and logging the results.
>
> |  | AutoPlan | ReAct (2 Shot) | ReAct (0 Shot) | Supervised |
> | --- | --- | --- | --- | --- |
> | Majority Votes (≥2 votes) | 80% | 73% | 71% | 91% |
> | Aggreements (all 3 annotators agree) | 90% | 93% | 92% | 95% |
>
> We used the majority vote from the three annotators for evaluation, and AutoPlan outperformed ReAct both with and without in-context examples. Furthermore, the annotators agreed with each other in over 90% of the cases.
>
> **Q6: Grammar errors distract from the paper’s findings.**
>
> A6: Thank you for your suggestion! We will invite a fluent English speaker to carefully review and edit the paper to eliminate any distracting grammatical errors.

---

### Official Review · Reviewer_gLwg · 2023-08-04

**Soundness:** 3

**Excitement:**

3: Ambivalent: It has merits (e.g., it reports state-of-the-art results, the idea is nice), but there are key weaknesses (e.g., it describes incremental work), and it can significantly benefit from another round of revision. However, I won't object to accepting it if my co-reviewers champion it.

**Paper Topic And Main Contributions:**

This paper studies the problem of using the LLMs model to come up with a sequence of high-level actions to solve the instruction following tasks. The paper proposes a new method, called AutoPlan, that consists of a "self-reflection" mechanism to modify the existing failure plans. The idea is that first LLMs come up with an initial plan, and then execute the actions in the environment, receiving the observations. Once the agent fails, it then does a "reflection" step that uses LLMs to summarize the failure and provide what to do to revise the previous plan. Based on the revision, the plan is updated. And the new plan is deployed in the environment again. Based on this repeated process, the plan is optimized, and the final plan is deployed.

Several results are provided in the paper. First, Table 3 shows that AutoPlan has better performance than ReAct under the zero-shot setting. for both ALFWorld and Hotpot QA.  Second, table 4 shows that the cost of AutoPlan is lower than the cost of React. Third, figure 3 shows ablate the components of batch size and the reflection step. The observation is that with larger examples, and training iterations, the model has better performance.

**Questions For The Authors:**

1. Do you keep the entire reasoning context in the context of the LLMs when generating the next action plan?

2. What are the failure cases?

**Reasons To Accept:**

1. The writing of the paper is decent but could be improved.  For instance, it is unclear to me whether the optimization process is done online or offline based on the writing in Section 3.

2. The performance of the model seems good, and ablation studies of the paper help to understand the method.

3. This method seems to be a nice combination of ReAct and Reflection.

**Reasons To Reject:**

1. I have one major concern about the method. It seems that the method runs as follows: given the target instruction, the paper runs AutoPlan on the simulator, and then finally deploys the final actions (reset the entire reasoning trace). I think this is "cheating" in the sense that you gather information from the environment, and use the feedback to try again, which means that this is NOT a zero-shot setting, but a trial-and-error approach until you succeed. For ReAct, it does not have a reset process, and it just keeps generating the next actions based on the observation. As a result, I DO NOT think this is a fair comparison. I think what the author should do is keep the entire reasoning trace in the context of the LLMs and see if it can solve the task.

2. Following the above discussion, this method needs a batch of data to run, which means that you need to provide a couple of data points to use.

3. Figure 3 seems to not have statistical significance. And what will happen if the batch size increases to 10 or 20?

4. Another issue is that the method is based on the assumption each state is "reversible", in the sense that the agent can go from one state to another state freely. If I want to deploy this method in a real robot, what will happen when the agent fails? Will the plan is able to recover the robot from failure cases?

**Reproducibility:**

3: Could reproduce the results with some difficulty. The settings of parameters are underspecified or subjectively determined; the training/evaluation data are not widely available.

**Reviewer Confidence:**

4: Quite sure. I tried to check the important points carefully. It's unlikely, though conceivable, that I missed something that should affect my ratings.

---

> ### Author Rebuttal · Authors · 2023-08-29
>
> **Q1: Online or offline optimization?**
>
> A1: Apologies for the confusion. The optimization process occurs in an online and off-policy manner. Specifically, we use the current plan to collect a batch of experiences (trajectories) and update the plan based on the reflection of these experiences. This will be clarified in the revised version.
>
> **Q2: Comparison setup for AutoPlan.**
>
> A2: To clarify, AutoPlan has distinct training and inference phases. During training, we use the current plan to collect experiences from the training set and update the plan based on reflections on these experiences. In the inference phase on test environments, AutoPlan does not reset and directly generates the action sequence in a single round. This is conditioned on the optimized plan and observations from test environments. The entire reasoning process occurs within the context of the LLMs. The only difference between ReAct and AutoPlan during inference is that ReAct uses in-context demonstrations, while AutoPlan uses the optimized task plan. This will be made clear in the revised version.
>
> **Q3: Extra data for training.**
>
> A3: AutoPlan does require training data, but it needs no more than 50 training environments for each dataset tested in order to generate a performant task plan.
>
> **Q4: Larger batch size.**
>
> A4: We further tested AutoPlan with a batch size of 8. The mean (standard deviation) of success rates are as follows:
>
> |  | Iteration 1 | Iteration 2 | Iteration 3 |
> | --- | --- | --- | --- |
> | Batch size 2 | 0.23 (0.32) | 0.4 (0.35) | 0.3 (0.3) |
> | Batch size 4 | 0.37 (0.32) | 0.53 (0.15) | 0.67 (0.15) |
> | Batch size 8 | 0.7 (0.2) | 0.67 (0.23) | 0.73 (0.15) |
>
> We were unable to test AutoPlan with a batch size of 16 due to the context size limitations of GPT-4. Larger batch size overall delivers better performance.
>
> **Q5: Significance testing of Figure 3.**
>
> A5: We conducted a t-test for each training iteration to compare batch sizes of 2 vs. 4 and 2 vs. 8. Batch size of 8 is significantly better than batch size of 2.
>
> |  | Iteration 1 | Iteration 2 | Iteration 3 |
> | --- | --- | --- | --- |
> | p-value (2 vs 4) | 0.44 | 0.35 | 0.013 |
> | p-value (2 vs 8) | 0.007 | 0.110 | 0.005 |
>
> **Q6: What are the failure cases and does the system have the capability to recover from them?**
>
> A6: The types of failure cases can vary, and a detailed error analysis is provided between lines 401-427. We will include the trajectories of typical failure cases in the appendix. For instance, in ALFWorld, the task might be to place a clean mug in the coffee machine. However, the LLM agent may mistakenly choose a cup, clean it, and place it in the coffee machine. Subsequently, it may engage in repetitive and meaningless actions.
>
> Regarding the system's ability to recover from failures, it largely depends on the nature of the failure. For example, if the task is to place an apple on a table and the apple is dropped, the LLM agent can instruct the robot to clean the apple and continue with the task, making adjustments for similar future scenarios. However, for irreversible failures, such as a broken apple, the system likely cannot recover without human intervention. This aspect will be further discussed in the revised version of the document.

---

### Official Review · Reviewer_hFiy · 2023-08-05

**Soundness:** 3

**Excitement:**

3: Ambivalent: It has merits (e.g., it reports state-of-the-art results, the idea is nice), but there are key weaknesses (e.g., it describes incremental work), and it can significantly benefit from another round of revision. However, I won't object to accepting it if my co-reviewers champion it.

**Paper Topic And Main Contributions:**

They propose AutoPlan, a prompting method to align LLMs to decision-making tasks. The method achieves similar performance with baseline with human-written demonstration.

**Reasons To Accept:**

1. The prompting method could transfer an existing langauge model into interactive agent tasks.
2. The author demonstrates great performance.
3. To stablize the tuning, the author propose to use trio reflection and large batch sizes.

**Reasons To Reject:**

1. The authors themselves did the human evaluation.
2. Compared with other baseline methods, AutoPlan needs extra tuning for the general task, while baseline such as AdaPlanner does not requires extra training time. Also, the author did not clarify what is the disadvantage of using in-context demonstration

**Reproducibility:**

3: Could reproduce the results with some difficulty. The settings of parameters are underspecified or subjectively determined; the training/evaluation data are not widely available.

**Reviewer Confidence:**

3: Pretty sure, but there's a chance I missed something. Although I have a good feel for this area in general, I did not carefully check the paper's details, e.g., the math, experimental design, or novelty.

---

> ### Author Rebuttal · Authors · 2023-08-29
>
> **Q1: Human Evaluation**
>
> A1: Thank you for the suggestion! We conducted an additional human evaluation of HotpotQA with three external human annotators. The full annotation instructions will be included in the paper's appendix. The annotation process involved four phases: reviewing the data points, checking the accuracy of predictions, verifying the consistency with supporting facts, and logging the results. The results are as follows:
>
> |  | AutoPlan | ReAct (2-Shot) | ReAct (0-Shot) | Supervised |
> | --- | --- | --- | --- | --- |
> | Majority Votes (≥2 votes) | 80% | 73% | 71% | 91% |
> | Agreements (all 3 annotators agree) | 90% | 93% | 92% | 95% |
>
> We used the majority votes from the three annotators as the evaluation results. AutoPlan still outperforms ReAct, whether or not in-context demonstrations are used. Additionally, we computed the level of agreement among the annotators and found that in over 90% of cases, all annotators were in agreement.
>
> **Q2: Extra Tuning for AutoPlan versus AdaPlanner**
>
> A2: AutoPlan does require a training phase to obtain the task plan. However, during inference, AutoPlan directly generates the action sequence based on the optimized plan and observations. In contrast, baselines like AdaPlanner and Reflexion involve a test-time self-reflection process to iteratively refine the action sequence. This introduces an additional inference cost, as demonstrated below in the ALFWorld (unit: US dollar):
>
> |  | AutoPlan | ReAct | AdaPlanner | Reflexion (5 trials) |
> | --- | --- | --- | --- | --- |
> | Inference Cost per Case | 1.6 | 3 | 2.1 | 17 |
> | Training Cost (24 cases) | 43 | N/A | N/A | N/A |
>
> **Q3: Disadvantages of Using In-Context Demonstrations**
>
> A3: On one hand, using in-context demonstrations increases the inference cost, as shown above and in Table 4 of the paper. On the other hand, generating these demonstrations requires additional human effort to annotate correct action sequences based on given environments and reward functions. AutoPlan, in contrast, only needs the latter two (environment and reward function) to operate effectively.

---

### Meta-Review · Area_Chair_kM6L · 2023-09-23

**Recommendation:** 3

**Metareview:**

Reveiwers have reached consensus in their reviews. While the result are reasonable, they are not conclusive in setting out the advatages of AutoPlan, the presenation can be improved to highlight the value of the approach, i.e. exanding on Table 4 in the paper. The paper should be updated with the results in the rebutal.

---

### Decision · Program_Chairs · 2023-10-07

**Decision:**

Accept-Findings

**Comment:**

Reveiwers have reached consensus in their reviews. While the result are reasonable, they are not conclusive in setting out the advatages of AutoPlan, the presenation can be improved to highlight the value of the approach, i.e. exanding on Table 4 in the paper. The paper should be updated with the results in the rebutal.